# Impact of Dye Encapsulation in ZIF-8 on CO_2_, Water, and Wet CO_2_ Sorption

**DOI:** 10.3390/molecules28207056

**Published:** 2023-10-12

**Authors:** Aljaž Škrjanc, Mojca Opresnik, Matej Gabrijelčič, Andraž Šuligoj, Gregor Mali, Nataša Zabukovec Logar

**Affiliations:** 1Department of Inorganic Chemistry and Technology, National Institute of Chemistry, Hajdrihova 19, SI-1001 Ljubljana, Slovenia; aljaz.skrjanc@ki.si (A.Š.); mojca.opresnik@ki.si (M.O.); matej.gabrijelcic@ki.si (M.G.); andraz.suligoj@ki.si (A.Š.); 2Postgraduate School, University of Nova Gorica, Vipavska 13, SI-5000 Nova Gorica, Slovenia; 3Faculty of Mathematics and Physics, University of Ljubljana, Jadranska ulica 19, SI-1000 Ljubljana, Slovenia; 4Faculty of Chemistry and Chemical Technology, University of Ljubjana, Večna pot 113, SI-1000 Ljubljana, Slovenia

**Keywords:** zeolitic imidazolate frameworks, water sorption, CO_2_ capture, organic dyes incorporation

## Abstract

The fast adsorption kinetics of zeolitic imidazolate frameworks (ZIFs) enable a wide range of sorption applications. The most commonly used framework, ZIF-8, is relatively non-polar. Increasing the polarity of ZIF-8 through the encapsulation of different polar species shows promise for enhancing the sorption performance for pure CO_2_. Recently, the outlook has re-focused on gas mixtures, mostly in the context of post-combustion CO_2_ capture from wet flue gasses. While water is known to sometimes have a synergistic effect on CO_2_ sorption, we still face the potential problem of preferential water vapor adsorption. Herein, we report the preparation of three ZIF-8/organic dye (OD) composites using Congo red, Xylenol orange, and Bromothymol blue, and their impact on the sorption properties for CO_2_, water, and a model wet CO_2_ system at 50% RH. The results show that the preparation of OD composites can be a promising way to optimize adsorbents for single gasses, but further work is needed to find superior ZIF@OD for the selective sorption of CO_2_ from wet gas mixtures.

## 1. Introduction

Zeolitic imidazolate frameworks (ZIFs) are a subgroup of metal organic frameworks (MOFs) that are formed from tetrahedrally coordinated imidazole and benzimidazole ligands to transition metals, forming 2D and 3D coordination networks [1]. The tetrahedral coordination and the similar bond angles to those observed in zeolites give the materials their name. Due to the framework’s similarity to zeolites, topology nomenclature from the zeolite field is used, with the most common ZIFs exhibiting the SOD [2,3,4,5], RHO [6,7,8], and LTA [9,10] topologies. The broad variety of linkers also led to the formation of ZIFs with non-zeolite topologies, such as lcs [4], cag [11], etc.

The most studied and widely used ZIF is the zinc 2-methylimidazolate with sodalite topology named ZIF-8 (Figure 1) [5]. Its widespread use as a functional material [3,12,13,14], can be attributed to its high chemical and thermal stability [15] as well as to the ease of its preparation [16]. Despite its high stability and fast sorption kinetics, ZIF-8′s CO_2_ uptake is relatively low at working temperature and pressure conditions (~1 atm, 298 K). The framework’s nonpolar linkers (Figure 1) provide no additional interaction with CO_2_ and, as such, the uptake is largely dependent on the surface area [17] associated with the synthesis method chosen [18].

To use the advantage of ZIF-8 stability and, at the same time, improve the CO_2_ sorption performance, various techniques have been applied to try to identify the adsorption sites, interactions, and adsorption mechanisms of ZIF adsorbents [19,20,21,22,23]. What has successfully been proven is that the metal node is not an adsorption site and CO_2_ sorption is attributed to week intermolecular interactions between CO_2_′s dipoles and the aromatic imidazole rings. Furthermore, the benchmark adsorption of ZIF-8 of around 1 mmol CO_2_ per g at 1 bar and 298 K [24] is low, if compared to its less stable structural analogues like ZIF-90 and ZIF-94, which exhibit an uptake of ~2 mmol CO_2_ per g at the same temperature and pressure. The difference is attributed to the presence of polar aldehyde groups on ZIF-90 and ZIF-94 linkers, since the metal modes and imidazolate rings are the same in both structures.

Throughout the years, different approaches towards adding functionality and thus increasing the CO_2_ uptake of ZIF-8 have been developed, resulting in three main synthesis techniques. The first two add functional groups either through linker modification, as mentioned in the previous paragraph [25,26,27,28], or through pore filling with dyes [29] or ionic liquids [30,31,32]. The third approach is defect creation [33,34,35]. In all cases, added functionality increases the interactions between the porous material and CO_2_ and enhances the uptake, with the exception of encapsulation where, potentially, the pores can be overfilled [29] and the accessible surface-area-dependent part of the sorption is impacted.

With encapsulation showing promising results with pure CO_2_, now the outlook is also focusing on gas mixtures in order to evaluate the potential of selective CO_2_ capture, e.g., from wet streams. While water is known to sometimes increase CO_2_ sorption in certain MOFs [36], we still face the potential problem that is preferential water adsorption due to the dyes’ and ionic liquids’ high hydrophilicity. Herein, we report on the preparation of three ZIF-8/organic dye (OD) composites using Congo red (CR), Xylenol orange (XO), and Bromothymol blue (BB) (Figure 2). The ODs chosen were from the list of the most common organic dyes used as pH indicators [37], metal ion indicators [37], and as model pollutants for photocatalysis [38], with different sizes, shapes, and functional groups. After successful OD incorporation, the impact of the structure modification on the sorption properties for water, CO_2_, and a model wet CO_2_ system at 50% RH was investigated.

## 2. Results

### 2.1. Synthesis and Characterization

The prepared ZIF-8 samples were first analyzed with powder X-ray diffraction (PXRD) to confirm successful framework formation (Figure 3). The modified ZIF-8 materials all show the major diffraction peaks corresponding to the diffraction pattern of the original ZIF-8. These patterns indicate that the addition of dyes into the reaction mixture still allowed for framework formation without a significant loss to crystallinity. The samples after activation, i.e., heating under vacuum to remove volatile compounds from pores (denoted with letter “A” in Figure 3), were again checked with PXRD to confirm that no thermal decomposition of the framework occurred.

Dye incorporation was observed visually as all OD samples exhibited vivid color (Figure 4, left) if compared to the white parent ZIF-8. The vibrancy did not change even after washing with MeOH. After the successful preparation of ZIF-8 and the ZIF-8OD frameworks, scanning electron microscopy (SEM) was used to determine morphology and particle size distribution (Appendix A). SEM imaging (Figure 4) showed that, in all cases, nanoparticles of up to 100 nm in size were successfully prepared with unmodified ZIF-8, and the BB and XO ZIF-8 showed octahedral particles, while ZIF-8CRs particles were more rounded and slightly smaller when compared to the rest.

To test for potential interactions between the framework and the organic dyes (potential shifts peaks), and the quantification of the OD amounts ^1^H and ^1^H-^13^C CPMAS NMR, was performed on the parent and three dye-modified samples (Figure 5, Appendix A). Visible in the ^1^H NMR of all is the two broad peaks belonging to ZIF-8, as well as the additional narrow peaks, that are assumed to be assigned to unreacted linker and solvent. The 1H-13C CP MAS NMR spectra of the ZIFs shows three large peaks belonging to C1, C2, and C3 (Figure 1). Peaks belonging to the organic dyes could not be observed, as such, acid digestion ^1^H liquid NMR was carried out (Appendix A). All liquid spectra have the expected four peaks from Dymethil sulfoxide as the used solvent, H_2_O, C2, and C3, with integrals for C2 and C3 matching the expected ratio of 2/3. The ZIF-8CR spectrum had some issues with anisotropy in the aromatic region; nevertheless, clear, very weak singlet peaks were successfully assigned and integrated to determine the dye loading. The dye loadings were estimated to be around 1 wt% for ZIF-8CR and ZIF-8BB, respectively, and 2 wt% for ZIF-8XO. The loadings determined were an order of magnitude lower than those reported in the literature for the hierarchically porous ZIF-8RB system [39]. The difference can be attributed to the difference in synthesis procedure. The literature synthesis relies on the kinetic entrapment of dye due to the addition of triethylamine base, which induces the formation of an intermediate ZnO phase, as well as the formation of mesopores leading to hierarchical ZIF-8 structures.

### 2.2. Sorption Studies

Activated unmodified and modified samples were analyzed using N_2_ physisorption to determine the specific surface area, micropore and total pore volume, as well as pore size distribution (Figure 6, Table 1, Appendix A). The CO_2_, water, and 50% RH CO_2_ isotherms (the material was sequentially filled, first with water vapour and then with CO_2_) were measured to evaluate the uptake of single gas/vapour and their mixture (Figure 7, Table 1).

## 3. Discussion

The successful in situ incorporation of the organic dyes into the ZIF-8 framework was initially confirmed by the strong vibrant colors of the samples (Figure 4). Soaking the dried samples in methanol overnight did not significantly colour the solution or lead to a visible reduction in the colour vibrancy of the samples, which hints at the strong anchoring of the dye molecules to the support. Previous studies on the adsorption of the dye Rhodamine B (RB) in ZIF-8 [40] revealed that interactions between the organic parts of the framework and the dye can be observed as slight shifts of the peaks in the ^1^H-^13^C CPMAS spectra. While a shift was observed for ZIF-8CR and ZIF-8BB (Figure 5), in our case, it was smaller than reported for Rhodamine B and was only visible for the terminal ring C atoms, denoted as C2 in Figure 1. The small shift is not sufficient to state unambiguously that this is the locus of the preferential interactions of the framework with the dyes, especially since the additional shifts due to π–π stacking observed for Rhodamine B [40] are not found in our samples. This might be due to an order of magnitude smaller amount of OD in our case when compared to the previously reported data. Acid digestion liquid ^1^H NMR spectra were collected with very long relaxation times (5 s) and high numbers of scans (32) to hopefully obtain clear OD proton peaks. The obtained spectra mostly show the solvent and linker signals, with some small contributions peeking out from the baseline. Only doublet and singlet signals from the dyes could be identified; higher couplings and multiplets were too spread out and disappeared into the baseline. Still, partial assignation and integration allowed for the determination of mass loading, with all ZIF-8OD having a mass loading of around 1 w%. The nanoparticle nature of the prepared frameworks is apparent in the ^1^H spectra, showing two broad peaks of the ZIF-8 framework with some additional peaks observed in the ZIF-OD samples (Figure 5). While the deconvolution of spectra was carried out, the assignation of the peaks belonging to the protons of the OD and the further determination of dye loading could not be achieved. From these results, we could only speculate that dyes are stabilized in pores through electrostatic interactions.

This speculation was further checked using UV-Vis spectroscopy. First, the dyes’ anionic forms were investigated (Appendix A). In the case of XO and BB, the anionic form is expected, due to the intense colour shift [41,42]. But we could not be sure if the colour shift is due to XO coordinating to Zn or just due to the deprotonation of the dye [41]. Therefore, the UV-Vis spectra of XO and BB water solutions with added base, and of XO with zinc acetate, were recorded. A comparison of the UV-Vis spectra of ZIF-8 and dye solutions shows that the dye is in its deprotonated ionic form, in both the case of ZIF-8BB and ZIF-8XO. The adsorption minimum at around 250 cm^−1^ that can be observed in the case of ZIF-8XO and deprotonated XO leads us to conclude that the ZnXO complex does not form. Complex formation would lead to higher adsorption in the region around 250 cm^−1^ leading to a lack of local minimum. An inspection of the spectra of ZIF-8CR shows only slight changes in the spectrum of the dye in the framework or in the solution, leading to the conclusion that CR is held in the pores by weak intermolecular interactions, while BB and XO appear to interact through electrostatic forces.

Nitrogen physisorption (Figure 6, Table 1) showed that no significant reduction in specific surface area was observed for ZIF-8XO, which has a similar S_BET_ to unmodified ZIF-8. However, the ZIF-8BB and ZIF-8CR samples showed a slight decrease in S_BET_. The micropore volume of the prepared materials was smaller or larger in proportion to the S_BET_. The unmodified sample exhibited a sharp increase in the adsorbed volume at higher relative pressure, indicating mesoporosity, most probably interparticle mesoporosity. This is also indicated by the significantly larger total pore volume determined, if compared to the three ZIF-8OD samples. ZIF-8CR, despite its smaller particle size, S_BET_, and lower micropore volume, has the highest total pore volume of all three ZIF-8OD samples, which can be attributed to its higher interparticle mesoporosity. ZIF-8CR also has a visible increase in adsorbed amount at high P/P_0_, although this is slightly smaller than the unmodified ZIF-8.

While there was little change in N_2_ physisorption between samples, a significant difference was observed in CO_2_ and water uptake (Figure 7, Table 1). All CO_2_ isotherms still kept a similar line shape, with varying slopes. ZIF-8BB and ZIF-8XO had significantly higher CO_2_ uptake than the starting material ZIF-8, while ZIF-8CR had a similar uptake despite a 200 m^2^/g lower S_BET_. However, we were only able to increase CO_2_ uptake with dye incorporation by a maximum of 18%, which is significantly less than the reported 54% increase for ZIF-8RB [29]. On the other hand, Abdelhamid [29] showed that, in all cases, methylene blue (MB) reduced the uptake when compared to parent ZIF-8. The significantly lower CO_2_ uptake of ZIF-8CR could also be due to the partial steric blocking of pores and CO_2_ adsorption sites of the framework imidazolates, as evidenced by the slight shift in the C2 ^13^C peaks in the NMR spectra. On the other hand, the high uptake of ZIF-8BB can be attributed to the comparatively sterically smaller size of the dye and its non-planar nature, leading to the formation of new dipole-based adsorption sites in the pores without blocking the already existing adsorption sites of ZIF-8.

Water isotherms show the opposite behavior. In all cases, the ZIFs exhibit the same adsorption isotherm shape. While ZIF-8CR had the lowest CO_2_ uptake, it had the highest water uptake at 7.3 mmol, which is almost four times that of the unmodified ZIF-8. The other two ZIF-8OD only doubled the water uptake when compared to ZIF-8. The ZIF-8OD’s increased water adsorption was expected, as all OD have hydrophilic functionalities present in their molecular structures. While quadrupling the uptake of ZIF-8CR compared to the unmodified sample, it is still lower than in some hydrophilic ZIF frameworks, like ZIF-93 and ZIF-90, which can reach water uptakes of up to 20 mmol/g. Also of interest is the hysteresis observed in all three ZIF-8OD samples, the most pronounced being for ZIF-8CR. The desorption isotherm of the latter exhibits relatively small hysteresis until 40% RH; however, after that point, the desorption curve reaches almost a plateau. This indicates that water almost irreversibly bonds in ZIF-8OD structures. The shift in adsorption is also apparent in the kinetics of the adsorption (Appendix A) in the BB and XO samples, with a slower adsorption up to 30% RH, after which the mass stabilizes significantly faster. Interestingly, the same was not observed for ZIF-8CR where adsorption steps took a similar amount of time throughout the RH region.

To test the possible interference or synergistic effects of water on CO_2_ sorption in advance, a model system was developed with 50% RH. Due to instrumentation limitations, a direct 50% RH CO_2_ isotherm was not possible. To work around the problem, a model protocol was developed in which water was first added to the system to a pressure where 50% RH would be achieved at 760 torr. After the addition of the water into the system, a CO_2_ isotherm was measured in the same manner as before for dry CO_2_. Due to the difference in hydrophilicity between the samples, both a full isotherm and a corrected isotherm, where the initial water uptake was subtracted, were plotted (Figure 7). The 50% RH isotherms (Figure 7, Table 1) showed that the expected problem of preferential water adsorption may be an issue in selected ZIF-8 OD systems, as for all three OD ZIFs a corrected CO_2_ uptake is lower than the one observed for ZIF-8. Furthermore, the CO_2_ uptake is the lowest for ZIF-8CR, where the water uptake was the highest despite a change in the adsorption curve from the one for dry CO_2_. From these, we can conclude that the water in our ZIF-8 OD is preferentially adsorbed and prevents CO_2_ sorption, therefore no synergistic effects could be determined. While desorption has a negative hysteresis, this is mainly due to the instrumentation error, where we cannot control the humidity during desorption and, as such, the hysteresis is partially due to resulting changes RH, not only pressure. Based on the available setup and the selected protocol, we could not evaluate the possibility that some sites, already saturated by water vapor, might allow competitive CO_2_ and water vapor sorption, if both CO_2_ and H_2_O are introduced in the system simultaneously. The selected protocol, therefore, revealed only the “worst-case-scenario” regarding CO_2_ uptake and allows, at best, for a rough estimation of the behavior of these materials in wet CO_2_ conditions. In general, for all three studied ZIF-8OD samples, the single water and CO_2_ uptakes are worse that in ZIF-8’s closest hydrophilic analogues, ZIF-90 and ZIF-94. However, the difference occurs for wet CO_2_ isotherms; in the case of ZIF-8OD the adsorption sites do not fully saturate with water and CO_2_ isotherms can still be collected. While hydrophilic ZIFs usually show promise using breakthrough curve measurements [43], on the other hand, in the field of MOFs, several materials were already identified, the most recent being the zinc triazolate oxalate framework CALF-20, which exhibits high CO_2_ uptake in humid conditions [44]. Our results and data from the literature thus indicate that more work must be carried out on understanding and developing alternative systems that are robust and selective towards not only CO_2_/N_2_, but also CO_2_/H_2_O. Ideally, we could point towards the preparation of ZIFs with hydrophobic linkers to prevent water uptake, but which include strong polarized bonds to increase the interactions of CO_2_ with the framework. This could lead to higher uptakes of CO_2_, even in humid conditions. One such potential avenue is in the sodalite phases of zinc 4,5-dichloroimidazolate [45,46].

## 4. Materials and Methods

### 4.1. Materials

Methanol (MeOH, >99%), (GVL, 99%), 2H-methylimidazole (MIM, 97%), Xylenol Orange disodium salt (XO), Bromophenol Blue sodium salt (BB), Congo red (CR) and zinc nitrate hexahydrate (Zn(NO_3_)_2_∙6H_2_O, 99%), and deuterium chloride (DCl (35%)/D_2_O) were purchased from Sigma Aldrich (Darmstadt, Germany); deuterated dimetyhlsulphoxide (DMSO-d6) was purchased from Eurisotop (Saint-Aubin, France) and was used without further purification.

### 4.2. ZIF-8 and ZIF-8 OD Synthesis

A solution of 1.51 g (4.6 mmol) zinc nitrate hexahydrate in 50 mL of MeOH was added to a solution of 0.15 g OD and 3.33 g (4 mmol) MIM in 50 mL MeOH. The combined solutions were left to stir covered for 24 h then centrifuged at 6000 rpm. The precipitate was then resuspended in fresh MeOH and centrifuged again. The washing with MeOH was repeated twice. The samples were then dried in a fan oven at 60 °C overnight. Samples for sorption experiments were activated in a static vacuum at 150 °C. ZIF-8 was prepared in the same way without the addition of OD into the linker solution.

### 4.3. Characterization

Powder X-ray diffraction data (PXRD) were recorded on a PANanalytical X’Pert PRO high-resolution diffractometer using CuK_α1_ radiation (1.5406 Å) in the 2θ range from 5 to 50° (100 s per step 0.033° 2θ) with a fully opened X’Celerator detector. The diffractograms were analyzed and the particle size calculated using the Sherrer equation with the HighScore Plus 4.9 program package (Malvern Panalytical B.V.).

Nitrogen physisorption isotherms were recorded at −196 °C using the Autosorb iQ3. Before the adsorption analysis, the samples were degassed under a vacuum for 10 h at 150 °C. The Brunauer–Emmett–Teller (BET) specific surface area was calculated from adsorption data in the relative pressure range from 0.005 to 0.02. The total pore volume (V_total_) was calculated from the amount of N_2_ adsorbed at P/P_0_ = 0.97 and micropore volume from t-plot (P/P_0_ = 0.15–0.3).

Liquid UV-Vis spectra were taken on an Agilent Cary 60 instrument equipped with 10 mm quartz cuvette. The optical properties of solid samples were taken using a diffuse reflectance mode on a Lambda 650 UV-vis spectrophotometer (PerkinElmer, Waltham, MA, USA), equipped with a Praying Mantis accessory (Harrick). The scan speed was 480 nm/min and the slit was set to 2 nm. Spectralon^®^ (Harrick Scientific Products, Pleasantville, NY, USA) was used for background correction.

Scanning electron microscope (SEM) images were taken using a Zeiss Supra 35 VP microscope with an electron high tension voltage of 1.00 kV and Aperture Size 30.00 μm.

CO_2_ and H_2_O isotherms were collected on a Surface Measurements Systems DVS Vacuum system. The samples were first degassed at 110 °C in a vacuum with the turbo pump on for 3h, then left to cool under a vacuum for 3 h. The isotherms were collected at 30 °C in the pressure ranges P/P_0_ from 0 to 0.90 for water and 0 to 1.0 for CO_2_, with P_0H2O_ being 31.5 torr and P_0CO2_ 760 torr. Humid CO_2_ isotherms were collected by adding a first step where water vapor was added to the pressure where, at 760 torr, 50% RH would be achieved. The CO_2_ isotherms were then collected the same way as for dry CO_2_.

Solid-state MAS (magic angle spinning) NMR spectra were recorded on a Bruker AVANCE NEO 400 MHz NMR spectrometer equipped with a 4 mm CPMAS probe. Larmor frequencies of ^1^H and ^13^C nuclei were 400.14 MHz and 100.62 MHz. Sample MAS frequencies were 15 kHz for the measurements of ^1^H and ^13^C spectra. ^1^H spectra were recorded using Hahn echo pulse sequence of π/2 and π pulses with duration of 2.5 μs and 5.0 μs, respectively, 100 scans and delay between scans of 5 s. ^1^H-^13^C cross-polarization (CP) spectra were recorded using a ^1^H excitation pulse of 2.5 μs and contact time of 2.5 ms, 3000 scans with a delay between scans of 5 s. The shift axis in all the spectra was referenced using an external reference of adamantane.

Digestion liquid ^1^H NMR was recorded using an AVANCE NEO Bruker 600 MHz spectrometer at room temperature. Approximately 1.5 mg of each sample was digested in a mixture of DCl (35%)/D_2_O (0.1 mL) and diluted with DMSO-d6 (0.5 mL). Data analysis was performed using TopSpin 4.0.9 (Bruker) software.

## 5. Conclusions

Herein, we report on a study of the sorption properties of a series of organic dye (OD) ZIF-8 composites. While ODs have proven to generally increase both CO_2_ and water sorption, the latter by up to four times, the significant increase in hydrophilicity of the framework negates the increase in CO_2_ uptake achieved when adsorbing CO_2_ in a model 50% RH system. The results show that OD encapsulation in ZIFs could, indeed, be a promising route for increasing the CO_2_ sorption of already promising sorbents in dry conditions. Furthermore, the ZIF-8CRs quadruplication of the water uptake could, in the future, potentially be combined with already hydrophilic ZIF/MOF materials to further increase their water uptake, which could be beneficial for applications in water harvesting or thermochemical-sorption-based energy storage. The lack of a synergetic effect of larger amounts of water on CO_2_ affirms the need for further investigation into developing not only porous materials with increased CO_2_ uptake, but porous materials with high CO_2_/water selectivity for effective CO_2_ capture from flue gasses or other wet gas mixtures.

## Figures and Tables

**Figure 1 molecules-28-07056-f001:**
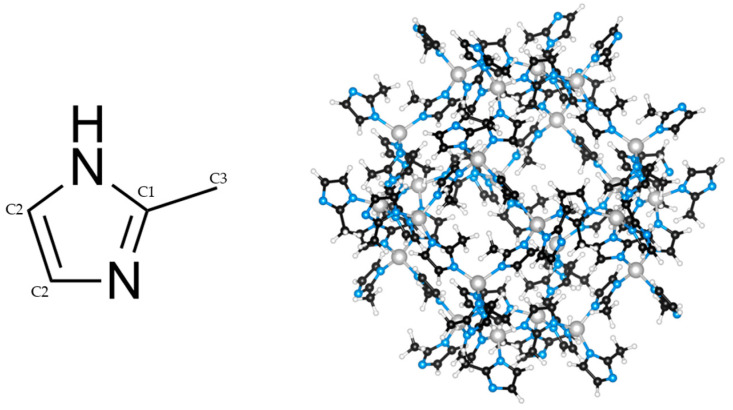
2-methylimidazole (**left**) and sodalite framework of ZIF-8 (**right**).

**Figure 2 molecules-28-07056-f002:**
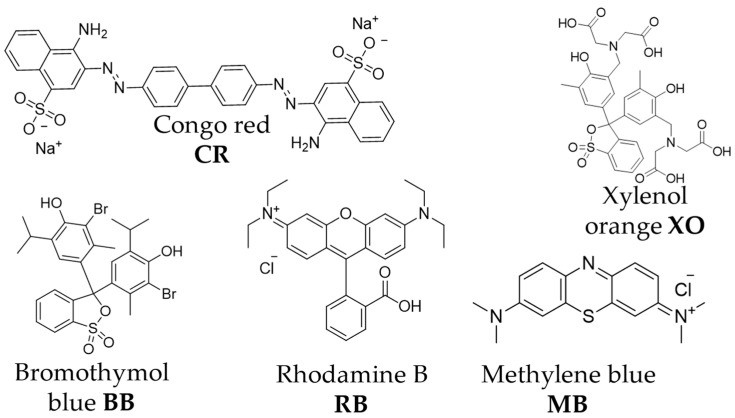
Structures of OD used for ZIF encapsulation in this study (CR, BB, XO) and in the literature (RB, MB, Ref. [29]).

**Figure 3 molecules-28-07056-f003:**
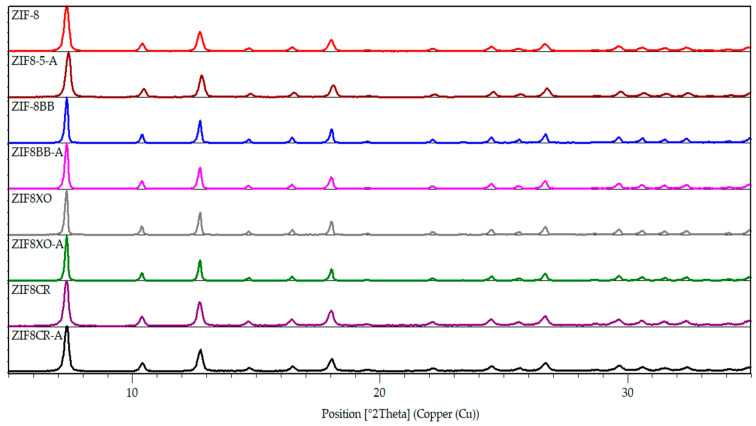
PXRD patterns of prepared ZIFs.

**Figure 4 molecules-28-07056-f004:**
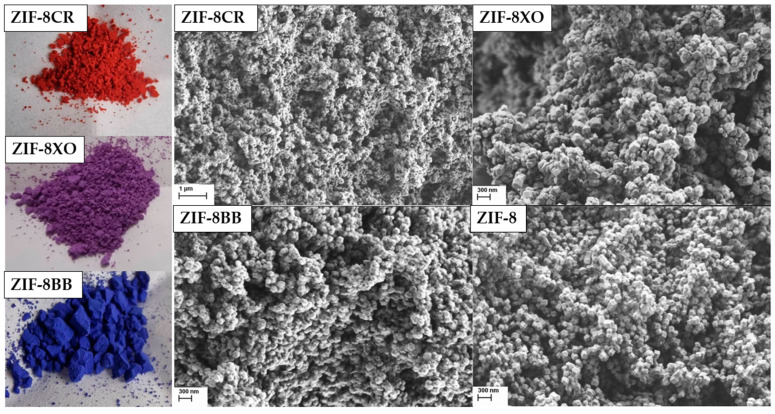
Photos of prepared ZIF-8OD (**left**) and SEMs of as synthesized samples (**right**).

**Figure 5 molecules-28-07056-f005:**
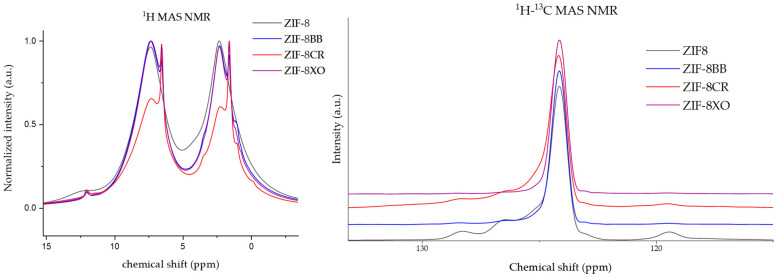
^1^H spectra of parent ZIF-8 and ZIF-8OD samples (left) and ^1^H-^13^C CPMAS spectrum around the peak at 124 ppm (right).

**Figure 6 molecules-28-07056-f006:**
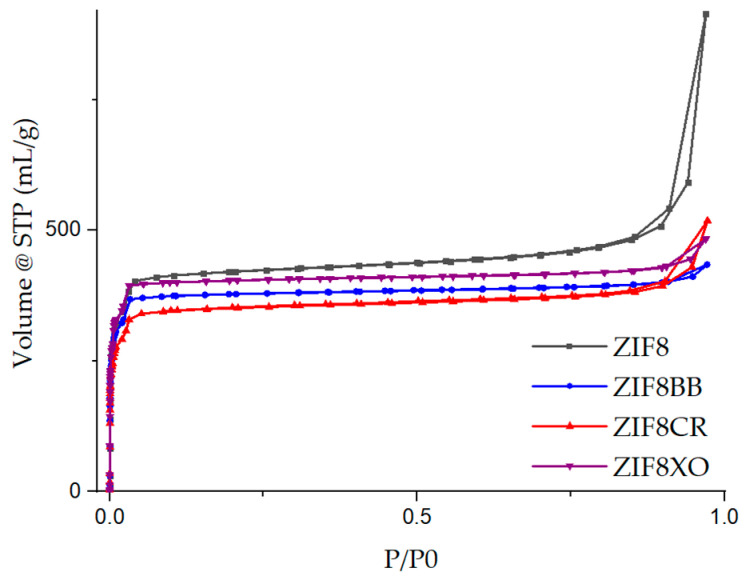
N_2_ physisorption isotherms of prepared samples.

**Figure 7 molecules-28-07056-f007:**
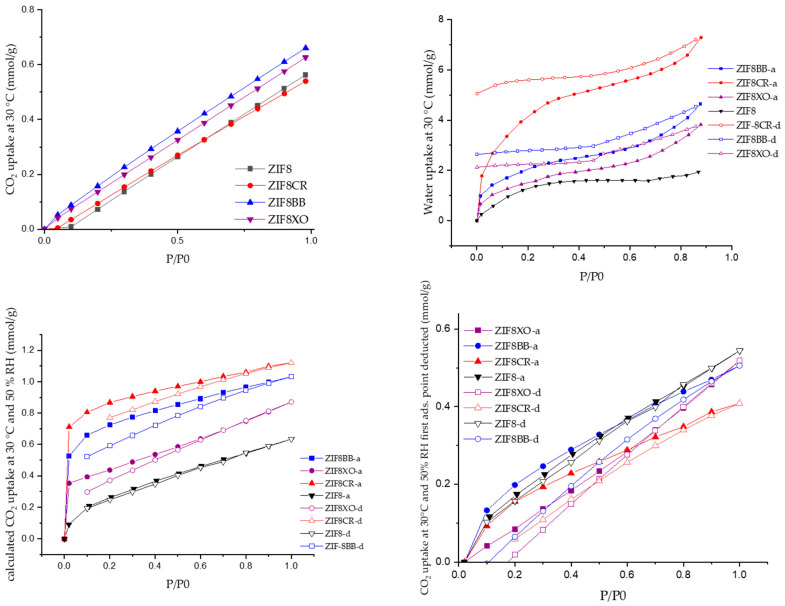
Dry CO_2_ and water isotherms (**top**), full 50% RH CO_2_ isotherm (**bottom left**) and CO_2_ isotherms corrected for water sorption (**bottom right**); Adsorption curves (full marks), desorption curves (empty marks).

**Table 1 molecules-28-07056-t001:** Results of sorption tests and data from the literature.

	N_2_ Physisorption			Gas/Vapour Uptake		
	S_BET_ [m^2^/g]	V_micro_ [mL/g]	V_total_ [mL/g]	CO_2_ [mmol/g]	H_2_O [mmol/g(wt%)]	Wet CO_2_ ^c^ [mmol/g]
ZIF-8	1566	0.59	1.41	0.56	1.9 (3.4)	0.54
ZIF-8BB	1457	0.56	0.67	0.66	4.6 (8.4)	0.51
ZIF-8CR	1314	0.50	0.80	0.54	7.3 (13.1)	0.41
ZIF-8XO	1560	0.59	0.75	0.62	3.8 (6.9)	0.52
ZIF-8RB ^a^	1000	0.32	0.62	0.79 ^b^	/	/
ZIF-8MB ^a^	400	0.12	0.27	0.41 ^b^	/	/

^a^ from Ref. [29], ^b^ measured at 25 °C, ^c^ after subtraction of the water adsorbed.

## Data Availability

Sorption data are available upon request.

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
