# Peer review of "Impact of Dye Encapsulation in ZIF-8 on CO2, Water, and Wet CO2 Sorption"

_molecules, 2023, doi:10.3390/molecules28207056_

Round 1

Reviewer 1 Report

The authors report the preparation and equilibrium adsorption performance of three organic-dye modified ZIF-8 materials. Although the characterization of the encapsulation is well evaluated, the sorption evaluation is limited and somewhat confusing. It is recommended that this work be published in Molecules only after significant revision of the introduction and sorption evaluation sections. Some comments to address these improvements are noted below.

1. Why were these particular dyes (CR, XO, and BB) tested for ZIF encapsulation?

2. Figure 7. The difference between the bottom plots is not clear. The caption description and the figure label for both bottom plots seem similar and the difference in sorbate cannot be easily understood. Re-labeling with more effective descriptors is recommended. Or replacing the y-axis with appropriate gas uptake label may be even better.

3. How is the “corrected isotherm” evaluated? What is the correction being made and why?

4. Sequential addition of H2O followed by CO2 may result in already saturated sorption sites that would otherwise be competitive for both CO2 and H2O.

5. Current sorption studies for “real gas mixtures” only characterize performance based on equilibrium measurements for adsorption. Both the desorption equilibrium at higher temperature, as well as the sorption kinetics are not evaluated here and must be addressed.

6. If the ZIF8-CR significantly improves H2O adsorption, would it make a more suitable candidate for dehumidification applications rather than for CO2 capture?

7. How does the water and CO2 uptakes of these ZIF-OD materials compare with typical CO2/H2O capture materials?

Minor editing required

Author Response

Reviewer 1

The authors report the preparation and equilibrium adsorption performance of three organic-dye modified ZIF-8 materials. Although the characterization of the encapsulation is well evaluated, the sorption evaluation is limited and somewhat confusing. It is recommended that this work be published in Molecules only after significant revision of the introduction and sorption evaluation sections. Some comments to address these improvements are noted below.

  1. Why were these particular dyes (CR, XO, and BB) tested for ZIF encapsulation?

The dyes were chosen from a selection of most common dyes used as pH indicators, metal ion indicators or as model pollutants for photocatalysis, with different sizes, shapes and functional groups. As Rhodamine B (RB), and Methyl blue (MB) were already investigated with ZIF-8, we decided to select the three that were not studied yet. Manuscript has been updated with an explanation on dye choice as follows:

“The OD chosen were from the list of most common organic dyes used as pH indicators[37], metal ion indicators[37] and as model pollutants for photocatalysis[38], with different sizes, shapes and functional groups.”

and updated Figure 1, where structures of RB and MB are added for comparison.

  1. Figure 7. The difference between the bottom plots is not clear. The caption description and the figure label for both bottom plots seem similar and the difference in sorbate cannot be easily understood. Re-labeling with more effective descriptors is recommended. Or replacing the y-axis with appropriate gas uptake label may be even better.

The plots have been renamed and the y axis updated as suggested.

  1. How is the “corrected isotherm” evaluated? What is the correction being made and why?

The »corrected isotherms« were plotted with the uptake of the first point deducted, as the calculation from weight % to mmol/g for two adsorbates with different molar masses can not be investigated together. Furthermore, it allowed for us to observe if the isotherm shape differs from the dry. Manuscript text has been updated to include isotherm shape differences to hilight this as follows:

»Due to the difference in hydrophilicity between the samples, both a full isotherm and a corrected isotherm, where the initial water uptake was subtracted, were plotted (Figure 7).”

  1. Sequential addition of H2O followed by CO2 may result in already saturated sorption sites that would otherwise be competitive for both CO2 and H2O.

We are fully aware of this potential problem, and have observed it for significantly more hydrophillic ZIFs. While we are aware of it, we are still limited by the capabilities of the equipment we have available (we can only measure the pressure, temperature and weight). While we could potentially add both vapours and gass into the sample cell, we are incapable of fully controlling how much of each is added and as such would have difficulty in achieving 50% RH. So as a compromise the approach where we have more controll was used. Still, a comment on this has been added in the manuscript:

»Based on available setup and selected protocol, we could not evaluate the possibility that some sites, already saturated by water vapor, might allow competitive CO2 and water vapor sorption, if both, CO2 and H2O, are introduced in the system simultaneously. The selected protocol therefore revealed only the “worst-case-scenario” regarding CO2 uptake and allows at best for a rough estimation of behavior of these materials in wet CO2 conditions.”

  1. Current sorption studies for “real gas mixtures” only characterize performance based on equilibrium measurements for adsorption. Both the desorption equilibrium at higher temperature, as well as the sorption kinetics are not evaluated here and must be addressed.

The sorption studies here were only preliminary to investigate the impact of the dyes on ZIFs, isotherms at other temperatures was not measured, due to still comparatively low adsorption when looking at other porous MOFs/ZIFs for CO2 sorption. Desorption plots were added where hysteresis was present (Figure 7). Relevant kinetics plots for water sorption were added and are discussed in the manuscript as follows:

“Of interest is also the hysteresis observed in all three ZIF-8OD samples, the most pronounced being for ZIF-8CR. The desorption isotherm of the later exhibits relatively small hysteresis till 40 % RH, however, after that point the desorption curve reaches almost a plateau. This indicates that water almost irreversibly bonds in ZIF-8OD structures. The shift in adsorption is also apparent in the kinetics of the adsorption (Figure S6) in the BB and XO samples, with a slower adsorption up to 30 % RH, after which the mass stabilizes significantly faster. Interestingly the same was not observed for ZIF-8CR where adsorption steps took a similar amount of time throughout the RH region. “

  1. If the ZIF8-CR significantly improves H2O adsorption, would it make a more suitable candidate for dehumidification applications rather than for CO2 capture?

While it significantly improves the water sorption, the water uptake is still somewhat low and lacks an inflection point, which is beneficial for most water sorption applications. Manuscript has been updated to emphasise this as follows:

“While quadrupling the uptake of ZIF-8CR compared to unmodified sample, it is still lower than in some hydrophilic ZIF frameworks, like ZIF-93 and ZIF-90, which can reach water uptakes of up to 20 mmol/g.”

  1. How does the water and CO2 uptakes of these ZIF-OD materials compare with typical CO2/H2O capture materials?

Manuscript has been updated with comparison to typical CO2 capture ZIF materials, as follows:

“In general, for all three studied ZIF-8OD samples, the single water and CO2 uptakes are worse that in ZIF-8 closest hydrophilic analogues ZIF-90 and ZIF-94. However, the difference occurs for wet CO2 isotherms; in the case of ZIF-8OD the adsorption sites do not fully saturate with water and CO2 isotherms can still be collected. While hydrophilic ZIFs usually show promise using breakthrough curve measurements[43]. On the other hand, in the field of MOFs, several materials were already identified, the most recent being the zinc triazolate oxalate framework CALF-20, which exhibit high CO2 uptake in humid conditions[44].”

Furthermore, we speculated on possible improvements in ZIF field for effective wet CO2 capture, as follows:

“Ideally, we could point towards preparation of ZIFs with hydrophobic linkers, to prevent water uptake, but which include strong polarized bonds to increase interactions of CO2 with the framework. This could lead to higher uptakes of CO2 even in humid conditions. One such potential avenue are sodalite phases of zinc 4,5-dichloroimidazolate[43,44].”

Reviewer 2 Report

Тhe manuscript is devoted to an interesting and particularly important topic. Тhe work will be improved if there is more discussion about the adsorption mechanism of individual molecules. It has been mentioned that the polarity of OD contributes to better adsorption, but this is not a sufficient explanation.

The phrase "leading to the formation of new adsorption sites in the pores without blocking the already existing adsorption sites of ZIF-8." - p. 7, line 157-158 needs additional clarification - which are the adsorption sites in the unmodified sample, which are new. What is their nature.

Two small notes:

-The size bars on Fig.4 are very small and difficult to be seen.

- p. 7, line 171 - "Table 3". There is no Table 3.

Author Response

Reviewer 2

Тhe manuscript is devoted to an interesting and particularly important topic. Тhe work will be improved if there is more discussion about the adsorption mechanism of individual molecules. It has been mentioned that the polarity of OD contributes to better adsorption, but this is not a sufficient explanation. The phrase "leading to the formation of new adsorption sites in the pores without blocking the already existing adsorption sites of ZIF-8." - p. 7, line 157-158 needs additional clarification - which are the adsorption sites in the unmodified sample, which are new. What is their nature.

The general literature consensus is that in most cases the adsorption sites for  CO2 are around the imidazole ring through intermolecular interactions. In this case CR interacting with the terminal part of the imidazole ring can be seen as blocking the adsorption site. We could not specificaly identify the new adsorption sites but literature tells us that dipole interactions strengthen CO2 physisorption, thus all polar bonds in the dyes can act as potential CO2 adsorption sites. Manuscript has been updated with this information:

“To use the advantage of ZIF-8 stability and at the same time improve the CO2 sorption performance, various techniques have been applied to try to identify the adsorption sites, interactions and adsorption mechanisms of ZIF adsorbents[19–23]. What has successfully been proven is that the metal node is not an adsorption site and CO2 sorption being attributed to week intermolecular interactions between CO2`s dipoles and the aromatic imidazole rings. Furthermore, the benchmark adsorption of ZIF-8 of around 1 mmol CO2 per g at 1 bar and 298 K[24] is low, if compared to its less stable structural analogues like ZIF-90, ZIF-94, which exhibit uptake of ~2 mmol CO2 per g at the same temperature and pressure. The difference is attributed to the presence of polar aldehyde groups on ZIF-90 and ZIF-94 linkers, since the metal modes and imidazolate rings are the same in both structures. “

Two small notes:

-The size bars on Fig.4 are very small and difficult to be seen.

Fig.4 bars have been modified for clarity.

- p. 7, line 171 - "Table 3". There is no Table 3.

Thank you, has been corrected.

Reviewer 3 Report

This is a work of a very high scienitic value. Despite the main conculsions are not somewhat glamorously "breathtaking", they are obviously important in terms of practical relevance and real application of MOF-based adsorbents. The paper is well written, adequate characterization and measurements were performed. A couple of additional notes and questions: 

1. It is recommended to determine a real OD content in solid composites. These data would provide a specific magnitude of the impact on the adsorption properties for such doping method. 

2. As described in the experimental, water vapor was let into the adsorption system up to 50% RH before CO2 adsorption experiment started. Are both water leaching and water condensation possible during subsequent CO2 isotherm recording? Has humidity been remeasured after the CO2 adsorption experiment? 

Author Response

Reviewer 3

This is a work of a very high scienitic value. Despite the main conculsions are not somewhat glamorously "breathtaking", they are obviously important in terms of practical relevance and real application of MOF-based adsorbents. The paper is well written, adequate characterization and measurements were performed. A couple of additional notes and questions: 

  1. It is recommended to determine a real OD content in solid composites. These data would provide a specific magnitude of the impact on the adsorption properties for such doping method. 

Acid digestion liquid 1H NMR was done and while the dye peaks in the 1H were very weak, we managed to integraTe some isolated singlets to determine the dye loading to be around 1w% for BB and CR and 2 w% for XO. Manuscript has been updated to include this information as follows:

»To test for potential interactions between the framework and the organic dyes (potential shifts peaks), and quantification of the OD ammounts 1H and 1H-13C CPMAS NMR was performed on the parent and three dye modified samples (Figure 5, S1 and S2). Visible in the 1H NMR of all is the two broad peaks belonging to ZIF-8, as well as additional narrow peaks, that is assumed to be assigned to unreacted linker and solvent. 1H-13C CP MAS NMR spectra of the ZIFs shows three large peaks belonging to C1, C2 and C3 (Figure 1). Peaks belonging to the organic dyes could not be observed, as such acid digestion 1H liquid NMR was done (Figure S3). All liquid spectra have the expected 4 peaks from Dymethil sulfoxide as the used solvent, H2O, C2 and C3, with integrals for C2 and C3 matching the expected ratio 2/3. The ZIF-8CR spectrum had some issues with anisotropy in the aromatic region, still clear very weak singlet peaks were successfully assigned and integrated to determine dye loading. The dye loadings were estimated to be around 1 wt% for ZIF-8CR and ZIF-8BB, respectively, and 2 wt% for ZIF-8XO. The loadings determined were an order of magnitude lower than those reported in literature for the hierarchically porous ZIF-8RB system[39]. The difference can be attributed to the difference in synthesis procedure. The literature synthesis rely on kinetic entrapment of dye due to the addition of triethylamine base, which induces the formation of intermediate ZnO phase, as well as the formation of mesopores leading to hierarchical ZIF-8 structures.”

  1. As described in the experimental, water vapor was let into the adsorption system up to 50% RH before CO2 adsorption experiment started. Are both water leaching and water condensation possible during subsequent CO2 isotherm recording? Has humidity been remeasured after the CO2 adsorption experiment? 

While we have experienced competitive adsorption in some other ZIFs in this case no leaching and condensation was observed. Humidity could not be measured after the adsorption experiment, as the we can only measure pressue, temperature and weight. We commented on the problem of sequential vs. simultaneous sorption, also as an answer to the comment of the first referee in the manuscript as follows:

 »Based on available setup and selected protocol, we could not evaluate the possibility that some sites, already saturated by water vapor, might allow competitive CO2 and water vapor sorption, if both, CO2 and H2O, are introduced in the system simultaneously. The selected protocol therefore revealed only the “worst-case-scenario” regarding CO2 uptake and allows at best for a rough estimation of behavior of these materials in wet CO2 conditions.”

Round 2

Reviewer 1 Report

The revisions and comments have addressed the original concerns. The revised manuscript is recommended to be published in Molecules.

Reviewer 2 Report

The manuscript is corrected according to the remarks and notes.